# Unintended Maladaptation: How Agritourism Development Policies in Iran Have Increased Vulnerability to Climate Change

Zabih-Allah Torabi [1,*] , Amir Reza Khavarian-Garmsir [2] , Colin Michael Hall [3,4,5,6,7]
and Neda Beiraghi Khatibi [1]

1 Department of Geography and Rural Planning, Tarbiat Modares University, Tehran 1411713116, Iran
2 Department of Geography and Urban Planning, Faculty of Geographical Sciences and Planning,
  University of Isfahan, Isfahan 8174673441, Iran
3 Department of Management, Marketing, and Tourism, University of Canterbury, Christchurch 8140, New Zealand;
  michael.hall@canterbury.ac.nz
4 The College of Hotel & Tourism Management, Kyung Hee University, Seoul 02447, Republic of Korea
5 Geography Research Unit, University of Oulu, 90014 Oulu, Finland
6 School of Business and Economics, Linnaeus University, 35195 Växjö, Sweden
7 Department of Service Management and Service Studies, Lund University, 22100 Lund, Sweden
* Correspondence: zabih.torabi@modares.ac.ir

**Abstract:** Implementing appropriate policies is crucial for adapting the agricultural sector to climate change. However, adopting incorrect policies can exacerbate unsustainable development. Hence, this study investigated the unintended consequences of agritourism development policies as a climate change adaptation strategy in the villages of Shahrud, Iran. It demonstrated how such policies have inadvertently heightened farmers' vulnerability to climate change impacts. Data were collected through 44 semi-structured interviews, which underwent thematic analysis to identify emerging patterns. The study's findings indicate that the rapid expansion of Agritourism in Iran, aimed at addressing climate change, has failed to achieve its intended goals. Inadequate government support, increased supply, legal gaps, and lack of empowerment were identified as contributing factors leading to unsustainable development and financial losses. Consequently, smallholder farmers were found to harbor negative perceptions of agritourism and expressed dissatisfaction with existing policies. These findings underscore the necessity of comprehensive policies and support systems to facilitate the effective implementation of sustainable agritourism by stakeholders in Iran.

**Keywords:** climate change; adaptation; agritourism policies; villages in Shahrud; Iran

## 1. Introduction

In the face of pressing global challenges such as climate change, the need for sustainable development practices has become increasingly urgent [1]. Within this context, agritourism, the integration of agriculture and tourism, has emerged as a promising avenue for fostering sustainable growth while addressing the impacts of climate change [2]. As agricultural systems face unprecedented challenges due to changing climate patterns, agritourism offers a pathway for farmers to adapt and thrive. However, the realization of the full potential of agritourism in the context of climate change adaptation relies heavily on effective policies [3,4].

Policies play a significant role in shaping the development of agritourism and its potential for climate change adaptation [5–7]. Effective policies can create a supportive environment that encourages the integration of agriculture and tourism, promoting sustainable practices and enhancing resilience to climate change impacts. By providing incentives and support, policies can facilitate the adoption of climate-smart agricultural techniques, such as sustainable water management, soil conservation, and biodiversity preservation [8,9]. These policies can also encourage the use of renewable energy sources

and promote the reduction in greenhouse gas emissions within agritourism operations. Furthermore, policies can facilitate the development of climate-resilient infrastructure, such as irrigation systems and farm diversification strategies, which can enhance the adaptive capacity of agricultural communities [10]. It is crucial for policymakers to consider the specific challenges and opportunities associated with agritourism in different regions, tailoring policies to local contexts and engaging stakeholders in the decision-making process [11]. By addressing the impacts of climate change and fostering sustainable agricultural practices through well-designed policies, agritourism can contribute to both economic development and environmental sustainability in rural areas [12,13].

Agritourism is a growing sector in Iran, offering an opportunity to experience the country's rich cultural and agricultural heritage. The country's diverse landscape and climate provide a vast array of agritourism opportunities, ranging from eco-tourism to farm stays. However, despite the industry's potential, there are also challenges facing the growth of agritourism in Iran. This country's efforts towards rural development have generally been unsuccessful in creating the necessary conditions for the development of tourism [14,15]. Despite previous efforts to address issues such as poverty and unemployment in rural areas through strategies such as wealth redistribution, infrastructure development, and basic service provision such as road and school construction, access to electricity, gas, and drinking water, significant challenges remain [1].

The economic difficulties facing small farms, climate change, pressure from international sanctions, and product price fluctuations have currently made smallholder farmers in Iran's rural areas abandon farms in the pursuit of new ventures [5,16,17]. As an alternative, smallholder farmers in Iran have adopted a multifunctional agriculture strategy as an attempt to diversify their activities [8,15]. This diversification of farm functions through the inclusion of entertainment and leisure activities also provides opportunities for agritourism, which has been suggested by many policy makers as a means of employment and economic diversification [18]. In addition, there is also interest by urban dwellers in consuming organic products produced in rural areas for both health benefits and to support and experience farm cultures [16,19]. In an effort to reduce the vulnerability of farmers to climate change and diversify their income, the Iranian government has recently prioritized agritourism. Despite promoting this type of tourism as a means of adaptation, it appears that unplanned and mandatory development may not achieve its intended goals and may instead cause serious issues for farmers.

This study addresses a significant theoretical gap by examining the specific negative consequences of agritourism development policies in the context of climate change adaptation. While agritourism is recognized as a potential strategy for enhancing economic and environmental resilience in rural communities, the unintended consequences of these policies remain understudied. The research focuses on the impact of agritourism development policies in the villages of Shahrud, shedding light on how they have increased the vulnerability of farmers to climate change impacts. Understanding these unintended consequences is vital for the development of comprehensive policies and support systems that effectively promote the adoption of agritourism by smallholder farmers in Iran. By filling this gap, the study contributes to scholarly literature and policy discussions, emphasizing the need to consider the unforeseen effects of climate adaptation policies and address the underlying drivers of vulnerability in rural areas.

The findings of this study have important implications for both the academic literature and policy practice, as they highlight the need to develop more comprehensive policies and support systems to enable the successful adoption of agritourism by smallholder farmers in Iran. Additionally, this study contributes to the broader conversation on the challenges of promoting sustainable development in rural areas, and the importance of addressing the underlying drivers of vulnerability to climate change impacts.

## 2. Background

The diverse agricultural and rural regions of Iran, enriched by varied climates, natural attractions, and local customs, hold great potential for rural and agritourism development [19]. Through systematic evaluation of rural and agricultural tourism and their strengths and weaknesses, these regions can promote sustainable rural growth and capitalize on existing economic, social, and environmental resources for the benefit of current and future generations [8]. Acknowledging the government's pivotal role in such contexts, given Iran's hierarchical structure wherein it acts as a developer, investor, and marketer, active and proactive engagement of the government is anticipated to serve as a significant catalyst for fostering sustainable rural and agricultural development through the promotion and facilitation of rural and agritourism. Based on this premise, we endeavored in this research to study the process of agritourism development in three selected villages in Shahrud county, taking into consideration the prominent role of the government in Iran (Figure 1) [20].

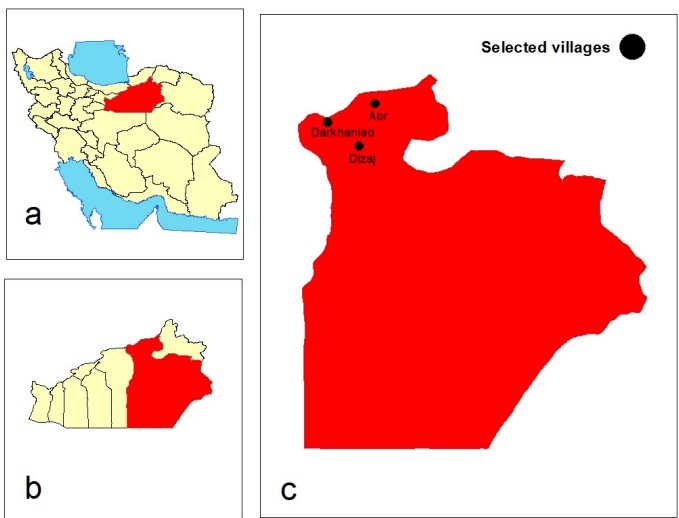

**Figure 1.** (**a**,**b**) shows the study area in Iran and Semnan province, respectively. (**c**) The location of the studied villages (Shahrud County) and the location of field data collection.

Situated within the Tehran–Mashhad International corridor, Shahrud, Iran, lies between two major population centers. This region, characterized by an altitude of 1400 m above sea level and diverse geographical climates, houses some of the largest wildlife sanctuaries and protected areas in the country (Iran's Cultural Heritage, Tourism, and Handicrafts Organization). As per the 2016 census, the city's population was 218,628, with 73,128 residing in rural areas, and approximately 42,009 individuals actively participating in agricultural activities, including 11,189 directly involved in farming (2016 census). Shahrud holds a prominent position in agricultural production within Semnan province and ranks among the 19 key agricultural centers in Iran [21]. Notably, its primary horticultural products are grapes and apricots, with the latter accounting for 10% of Iran's total output and over 1% of global production (Shahrud Agriculture Jihad). The region's agricultural products are harvested from mid-spring to early autumn, rendering it a potential attraction for agritourism throughout a significant portion of the year. Several successful agritourism projects, such as tourist gardens, have been established in the area. Additionally, urban dwellers' traditional purchases of rural products are prevalent. Presently, investments in multifunctional agriculture are gaining popularity in Shahrud, aimed at drawing a substantial number of tourists. For this study, we chose three villages in Shahrud that have recently experienced growth in agritourism and have been subject to investment: Dizaj, Abar, and Darkhaniab. These villages have captivated numerous tourists with their picturesque landscapes and diverse agricultural offerings. Furthermore, in response to

recent climatic changes, they are striving to enhance their income by improving the value chain of their agricultural produce.

## 3. Literature

Climate change is an issue that affects everyone on the planet, but it is especially challenging for rural communities whose livelihoods depend on natural resources such as agriculture, land, water, pasture, and forests [13]. In many developing countries, agriculture is the primary source of income for rural residents, and changes in weather patterns and other climate-related factors can significantly impact the productivity of crops [15,22]. As a result, adaptation to climate change has become a critical issue for rural development and the sustainability of agricultural systems [12,13].

Agricultural activities are particularly vulnerable to climate change [23]. Climate change can cause irregular rainfall patterns, more frequent and severe weather events, and changes in temperature that affect crop yields, pest and disease prevalence, and soil quality [24]. These changes can have cascading effects on the rural communities that depend on agriculture for their livelihoods [22]. As such, it is essential to find ways to adapt to climate change and reduce the risks of agricultural systems [25,26].

One promising strategy for adapting to climate change in rural areas is promoting multi-functional agriculture, which can help diversify income streams and reduce poverty and unemployment [8,9]. Agritourism is an excellent example of multi-functional agriculture, as it can potentially provide an additional source of income for farmers while also promoting sustainable agriculture, while educating the public about the impacts of climate change, and supporting adaptation strategies [10,17,25]. Agritourism can also help mitigate the impacts of climate change by promoting conservation and preservation of natural resources, such as soil, water, and forests [27,28].

Governments can play a crucial role in supporting agritourism as an adaptation strategy for climate change [11]. By providing funding and grants, regulatory support, promotion and marketing, and education and training, governments can help farmers develop sustainable agritourism businesses [10,29]. Funding and grants can help cover the costs of infrastructure improvements, marketing, and other expenses associated with starting an agritourism business [11]. Regulatory support can create guidelines that promote sustainable agritourism practices and ensure the safety of visitors [26]. Promotion and marketing can raise awareness of the local agricultural industry and encourage visitors [9,11]. Education and training can provide farmers and ranchers with the skills and knowledge needed to manage visitors and operate successful agritourism businesses [30]. One of the primary benefits of agritourism is that it can provide an opportunity for farmers to showcase their adaptation strategies to visitors [29]. For example, a farmer who has implemented water conservation measures in response to drought conditions can explain their methods to visitors, who may then be inspired to implement similar measures in their own communities [31]. By sharing their experiences with visitors, farmers can also learn from others and improve their own practices [9]. This knowledge exchange can help farmers and communities become more resilient to the impacts of climate change [26].

In addition to providing economic benefits to farmers and rural communities, agritourism can also help educate the public about the impacts of climate change on agriculture and the importance of sustainable farming practices [22]. By raising public awareness, agritourism can build support for policies and practices that promote sustainable agriculture and mitigate the impacts of climate change [10–12], while creating opportunities for people to connect with and learn from each other [32].

*Government and Agritourism*

Agritourism offers a range of economic and social benefits to rural communities [33,34]. However, the development of agritourism requires significant investment in infrastructure, marketing, and training, which can be challenging for small-scale farmers and rural communities [17].

Governments can play an important role in supporting the development of agritourism by providing various forms of support to farmers and ranchers [35]. The first way in which governments can support agritourism is by providing funding and grants to farmers who want to start agritourism businesses [36,37]. This can help cover the costs of developing infrastructure such as accommodation, restaurants, and attractions that are necessary to attract tourists [38]. Funding can also be used to improve marketing efforts and increase the visibility of the agritourism business, both locally and internationally [39]. Governments can also provide grants for training programs and workshops that help farmers acquire the necessary skills to operate agritourism businesses effectively [40,41]. Another way in which governments can support agritourism is by creating regulations and guidelines that promote sustainable agritourism practices and ensure visitor health and safety [17,42]. Regulations can also ensure that the agritourism industry operates in a sustainable manner, with minimal impact on the environment and natural resources [43]. This can help build the reputation of the agritourism business and attract more visitors over time [30].

Promotion and marketing are also crucial for the success of an agritourism business [44]. Governments can play a critical role in promoting agritourism by creating marketing campaigns and tourism development initiatives that showcase the region's agricultural heritage and attract visitors to the area [14,17,45,46]. Governments can also collaborate with local tourism boards and travel agencies to promote agritourism in the region, both domestically and internationally [8]. This can help raise awareness of the local agricultural industry and increase the demand for agritourism services [36,47].

Education and training are essential for the development of agritourism businesses [48,49]. Governments can provide education and training programs to farmers who are interested in starting an agritourism business [27,50,51]. This can help ensure that they have the necessary skills and knowledge to manage visitors and operate a successful agritourism business [52]. Training can cover topics such as visitor management, customer service, marketing, and financial management [5,53–55]. The government's support can provide significant benefits to rural communities, including increased economic opportunities, job creation, and sustainable agricultural practices [40,56,57]. By supporting the development of agritourism, governments can also help promote the importance of farming and food production to the public, contributing to the preservation of rural heritage and culture [43].

However, government interventions in climate change adaptation and agritourism development programs may fail due to a lack of planning and coordination, insufficient funding, lack of technical expertise, political interference, and a lack of participation from farmers or other stakeholders [58,59]. To avoid such failures, it is crucial to involve all relevant stakeholders in the design and implementation of these programs and to provide sufficient resources and technical expertise [60,61]. Therefore, governments should adopt a participatory and inclusive approach that involves all stakeholders, including farmers, civil society organizations, and private sector actors, to effectively adapt the agricultural sector to climate change and develop policies that support farmers [10,62]. Governments can also establish effective monitoring and evaluation systems to ensure that programs achieve their intended goals and have a positive impact on target communities [1,16,63]. Additionally, governments must allocate sufficient resources and technical expertise to climate change programs and prioritize agricultural development, especially in developing countries where the agricultural sector serves as the main livelihood for a significant portion of the population [10].

## 4. Methodology

Thematic analysis, a qualitative method, was employed to examine the challenges and issues arising from the rapid growth of agritourism in Iran. Respondents were selected using purposive and snowball sampling techniques, leveraging personal contacts to identify engaged individuals capable of providing pertinent and rich data [64]. The snowball sampling technique, as described by Torabi, Rezvani [65], involves existing participants introducing new ones, commonly utilized in qualitative research when accessing

informed participants is challenging due to limited target populations or sensitive research topics, as emphasized by Browne. This approach was favored to ensure representation of diverse perspectives, especially from those with unique, distinct, or crucial views on the tourism phenomenon. The study encompassed local villagers, including tourists, managers, and farmers, who directly or indirectly benefited from agritourism activities. A total of 44 respondents were included in the study (Table 1). Efforts were made to include knowledgeable respondents by focusing on the impact of climate change on agritourism's development specifically. This topic is sensitive and conservative managers in the public sector and traditional families may not share their opinions about their business unless they trust the researchers. The majority (68.18%) of the respondents were male, and the low participation of women in the research may be due to cultural reasons and the greater role of men in agritourism activities in the studied villages.

**Table 1.** Respondents' profile.

| Sector | Dizaj | Abr | Darkhaniab |
|---|---|---|---|
| Farmers | 10 | 7 | 9 |
| Tourists | 4 | 4 | 3 |
| Local government managers | 2 | 3 | 2 |
| Total | 16 | 14 | 14 |

*4.1. Data Collection*

During the spring and summer of 2022, data collection was carried out through pre-arranged appointments, offering participants the option of phone or in-person interviews. Potential respondents were contacted via phone and invited to take part in the survey. Two rural planning students, experienced in conducting interviews, were trained to collect primary data using a semi-structured interview questionnaire comprising seven questions related to climate change and agritourism, based on the existing literature. The interviewers received training on how to conduct interviews effectively, tailored to the unique circumstances of each village, and the approach required to engage local communities and gather essential information. Most interviews were audio-recorded, except for 11 respondents who opted not to have their voices recorded. For these cases, the interviewers took detailed notes. Data from three villages were collected in Farsi and later translated into English.

Interviews were conducted at locations chosen by the respondents, either their homes or workplaces, with durations ranging from 40 to 100 min. Prior to the interviews, respondents' consent was obtained, and the researchers maintained transparency and confidentiality in all discussions. The study's semi-structured interview guide consisted of four questions aimed at exploring the findings on agritourism in Iran, as presented in Table 2:

**Table 2.** Interview questions.

| | |
|---|---|
| 1 | What impact has the rapid growth of agritourism had on local markets? Please compare the situation before and after the development of agritourism, providing specific examples if applicable. |
| 2 | How has the government implemented agricultural development plans in rural areas of Shahrud to support agritourism? What challenges do farmers face in terms of empowerment programs, capacity building, and infrastructure incentives? |
| 3 | Has the government provided support for agritourism activities? If not, explain the repercussions of the lack of government support on farmers' training and preparation for participating in agritourism. Additionally, what minimum standards and training should be required for farmers entering the agritourism sector to ensure its success and sustainability? |
| 4 | In your opinion, have the principles of sustainability been overlooked in the development of agritourism? If so, elaborate on how current business practices in agritourism in Iran deviate from sustainable principles, particularly concerning land use preservation, indigenous culture, and agricultural productivity. |

The research process continued until theoretical saturation was achieved, resulting in 44 interviews. The research team determined that further interviews would not yield additional insights.

*4.2. Data Analysis*

This study utilized an analytical framework to analyze interview data through thematic analysis. Verbatim transcriptions of recorded interviews and notes were reviewed multiple times for familiarity and to enhance transcription reliability [66]. The immersion technique, which involves re-reading study materials, was used to reveal underlying meanings, constructs, themes, and patterns. MAXQDA, a textual analysis program, was utilized during the analysis process to supplement the thematic framework development. Logical and intuitive thinking were employed to make judgments about the importance of issues, meaning, and implicit connections between ideas to address research questions fully. Open coding and charting were also utilized to visualize data by considering theme familiarity, where data were coded according to research queries, and interesting features of data were identified. Semantic and conceptual relationships were used to group codes, and each group was given a name. A network of code groups was created, and main themes were selected for each network, with other sub-codes/sub-themes linked to the main themes to better understand the relationships between concepts.

To ensure research validity, transcribed interviews were sent back to respondents for review to ensure content and interpretation accuracy. However, only one-third of respondents provided additional feedback on their interviews. The emergent thematic structure and codes used during analysis ensured that the social distribution of issues raised were captured and reflected the subjectivities of the experiences shared by respondents.

**5. Results and Discussion**

*5.1. Climate Change, Government's Insufficient Support, Regulations and Ignorance*

The analysis of the interview data resulted in findings that indicate that the government shows support for the development of agritourism in announcements and media as a means of adapting to climate change, and is recognized as the primary implementer of agricultural development projects in rural areas of Shahrud. One of the farmers responded to the question about his decision to combine tourism with agricultural activities, stating that *"his land is approximately three hectares, and the family's expenses are steadily increasing. However, the income generated from the land does not suffice to meet their needs. Furthermore, his small orchard has been affected by frost for about three years, resulting in a lack of significant harvest. Consequently, with the support of government facilitators, he chose to engage in tourism, selling his products to tourists and occasionally providing accommodation on his farm."*

However, it was observed that farmers have not been adequately supported in this area, particularly in terms of empowerment programs, capacity building, and incentives for infrastructure reconstruction.

The interviewees asserted that public policies could effectively contribute to the setting up of groundwork for promoting entrepreneurial culture in agritourism through building capacity and empowering local communities. In this regard, experiences of countries like China [67], the United States [68], and Australia [69] demonstrated that the government's financial, educational, and supervisory support for agritourism entrepreneurs is necessary to launch such initiatives. Government support in Iran is even more important, given that this country does not have a strong private sector or NGOs, with almost all economic and social activities being either governmental or government-oriented [21]. Therefore, the possibility of success for development plans in Iran is slim without strong financial and governance support from the public sector.

The interview data indicated that many farmers are engaged in agritourism activities without any special training due to lack of sufficient government support. With regard to the engagement of unskilled people in agritourism, one of the local managers said: *"Many villagers have prepared their farms and houses to accommodate tourists, but the majority of them lack*

*hospitality and marketing skills. I believe one of the reasons for these farmers' failure is their instant and unskilled entry (into agritourism). People who do not possess a minimum set of standards should not be allowed to get engaged in agritourism."*

In this regard, one of the farmers expressed that, "In general, the government has not provided specific support to us. If we were to establish rural accommodations, we could have received more substantial support from the government because the support process was officially recognized. However, in this endeavor, we were constantly caught between the Ministry of Agriculture and the Ministry of Tourism, with no clear authority overseeing the development of agritourism. All promotional organizations launched agritourism initiatives, but when it came to providing support, they all stepped back."

Moreover, the participants contended that the available regulations do not provide any special support for farmers (as service providers) or tourists (as service recipients). With no clear definition of agritourism in Iran's regulations and official documents, this builds a complex terrain where such activities can be hard to classified, and as such agricultural tourists' and farmers' rights are neglected in governmental services, insurance, related rules, and security issues. In other words, available rules do not support agritourism. One of the seasoned farmers who endured hardships to receive his agritourism license claimed: *"When I started agritourism business, I discovered that I could not receive insurance because this job is not officially acknowledged in the administrative system of Iran Moreover, I could not receive an officially recognized license to start my business. That's why most of the agritourism businesses do not have any license and are unofficial. In time of crisis, the host or tourists do not receive any support."*

Based on the interviews conducted, it was found that public policies can play a vital role in fostering an entrepreneurial culture in the field of agritourism. This can be achieved by enhancing the capacity and empowering local communities. Various countries' experiences have shown that providing government support to farmers is essential for them to adapt to the effects of climate change, as demonstrated in studies conducted by Ammirato, Felicetti [32], Karampela and Kizos [70], and MacKay, Nelson [71]. Our findings also indicate that development programs are unlikely to succeed without adequate financial support and strong governance from the public sector.

The lack of government support has resulted in many farmers engaging in agritourism activities without any special training, which can lead to a lack of necessary skills and standards [8]. Without clear definitions and regulations for agritourism in Iran, it becomes challenging to classify such activities, resulting in neglected rights for both farmers and tourists in governmental services, insurance, related rules, and security issues. To address these issues, the government needs to provide financial, educational, and supervisory support for agritourism entrepreneurs, including capacity building and infrastructure incentives [36]. Proper training and regulation of agritourism activities are also necessary to ensure that farmers possess the necessary skills and meet minimum standards [32]. By providing the necessary financial, educational, and supervisory support, the government can potentially create an environment conducive to agritourism development while minimizing its negative impacts [30,48,51,72].

*5.2. Unsustainable Agritourism*

The lack of new approaches and growth in agritourism has led to incompatibilities between business practices and the principles of sustainable agritourism and community resilience in Iran. The researchers from [67] believe that the income generated as a result of farmers' involvement in agritourism supplements rather than replaces the one earned through production activities. One of the participants criticized the development of agricultural tourism, stating: *"We expected that by combining agricultural and tourism activities, we could increase agricultural productivity in small-scale lands and mitigate the effects of climate change. Additionally, we anticipated that these activities would promote the rural culture of the region. However, after a few years since the beginning of agricultural tourism activities, we have*

*not achieved our goals. One of the reasons for this is rapid and unplanned growth, which led us to neglect education and culture in promoting the principles of sustainable development.*"

However, in Iran, production-related activities are marginalized as a result of farmers' involvement in agritourism. In fact, sustainable agritourism may prevent land use change, preserving originality and indigenous culture, and boosting agricultural productivity [68]. Nonetheless, the executed plans in Shahrud County have mainly concentrated on tourists' leisure and entertainment at the expense of traditions, environment preservation, and promotion of product quality—instead of quantity. In this regard, one of the local managers said:

"The competition of attracting tourists and selling agricultural products to them is so intense that the farmers resort to anything to increase their income . . . Generally, it can be claimed that sustainable development goals are not known (to the local people) and most of the agritourism activities are carried out under no supervision because agritourism is not officially recognized . . . Also, instead of focusing on product efficiency in agritourism, most of the farmers prioritize entertaining activities."

One of the tourists criticized the management situation on the farms and stated:

"I expected to experience a pristine agricultural environment, but parties were held on the farm, which was contrary to my expectations and caused long-term damage to the farm. Buildings had been constructed on the farm, and the presence of tables and chairs scattered all over had depleted the soil resources. In general, my impression was that other agricultural activities were not a priority on this farm."

To avoid these negative impacts, it is important for agritourism to be managed sustainably [3,4]. This may involve setting limits on the number of visitors, implementing water conservation and waste management practices, and ensuring that local communities and workers are treated fairly [2]. The findings of this study are consistent with previous research on sustainable agritourism. For example, Barbieri noted that sustainable agritourism requires a balanced approach that considers economic, social, and environmental sustainability. Paniccia and Baiocco [73] also emphasized the importance of sustainable management of agritourism to minimize its negative impacts. The study highlights the need for sustainable management of agritourism in Iran. The lack of new approaches, coupled with instant entry into agritourism, has led to incompatibilities between practices in this business and the principles of sustainable agritourism and community resilience.

### 5.3. Failure and Loss of Financial Resources

The study found that many farmers had entered agritourism with the goal of increasing their income and recovering lost capital but were unable to continue their activities due to various challenges. Some of the challenges cited by the farmers included a lack of government support, ignorance about how to run an agritourism business, and the absence of proper regulations. In response to the question of why farmers' investments had not yielded the desired outcomes, one of the managers stated: "*I believe that one of the most significant reasons for the failure of these programs was the lack of adherence to sustainability principles. In other words, we did not adequately educate the suppliers on how to conduct themselves in economic, environmental, and social dimensions. The rapid growth of the initiatives hindered the depth of the teachings. I think, at that time, the primary concern of the managers was to increase the number of people interested in establishing agricultural farms, rather than focusing on ensuring sustainable practices.*"

Many farmers reported losing their savings and described agritourism as a high-risk business with a slim success rate. The interviewees also expressed the belief that many tourists who had invested in agritourism had lost their minimal financial resources, leading to frustration and a lack of trust in new approaches. One of the farmers interviewed was particularly pessimistic about the future of agritourism, saying that the industry had actually caused them to lose money and that they regretted getting involved. "*Agritourism, which was supposed to boost our income, made us lose our savings; hence, we quit. Many of us invested a lot in changing our farms' landscape, reconstructing traditional houses, and planting*

*trees and new products. I wish we had not entered agritourism or were engaged in other activities with lower risks."*

The study found that many smallholder farmers engaged in agritourism had lost their savings and described the industry as high-risk with a slim success rate. This sentiment was echoed by the interviewees, who expressed frustration and a lack of trust in new approaches. One farmer interviewed even regretted getting involved in agritourism and wished they had engaged in other activities with lower risks. One of the participants discussed the effects of farmers' failure in tourism activities, stating: *"Disappointment with tourism can have detrimental effects on the development of this industry. Although tourism can be highly profitable in certain circumstances, if you were to ask any farmer now to invest in this industry, they would undoubtedly refuse. Overall, I believe that restoring trust will require a significant amount of time."*

The study provides important insights into the challenges faced by smallholder farmers engaged in agritourism. The lack of government support, ignorance about how to run an agritourism business, and the absence of proper regulations were identified as significant barriers to success in agritourism [74–76]. Policymakers and industry stakeholders should consider these findings when designing policies and initiatives aimed at supporting smallholder farmers in the agritourism sector [77]. The converging experiences of the interviewees are summarized and mapped in Figure 2.

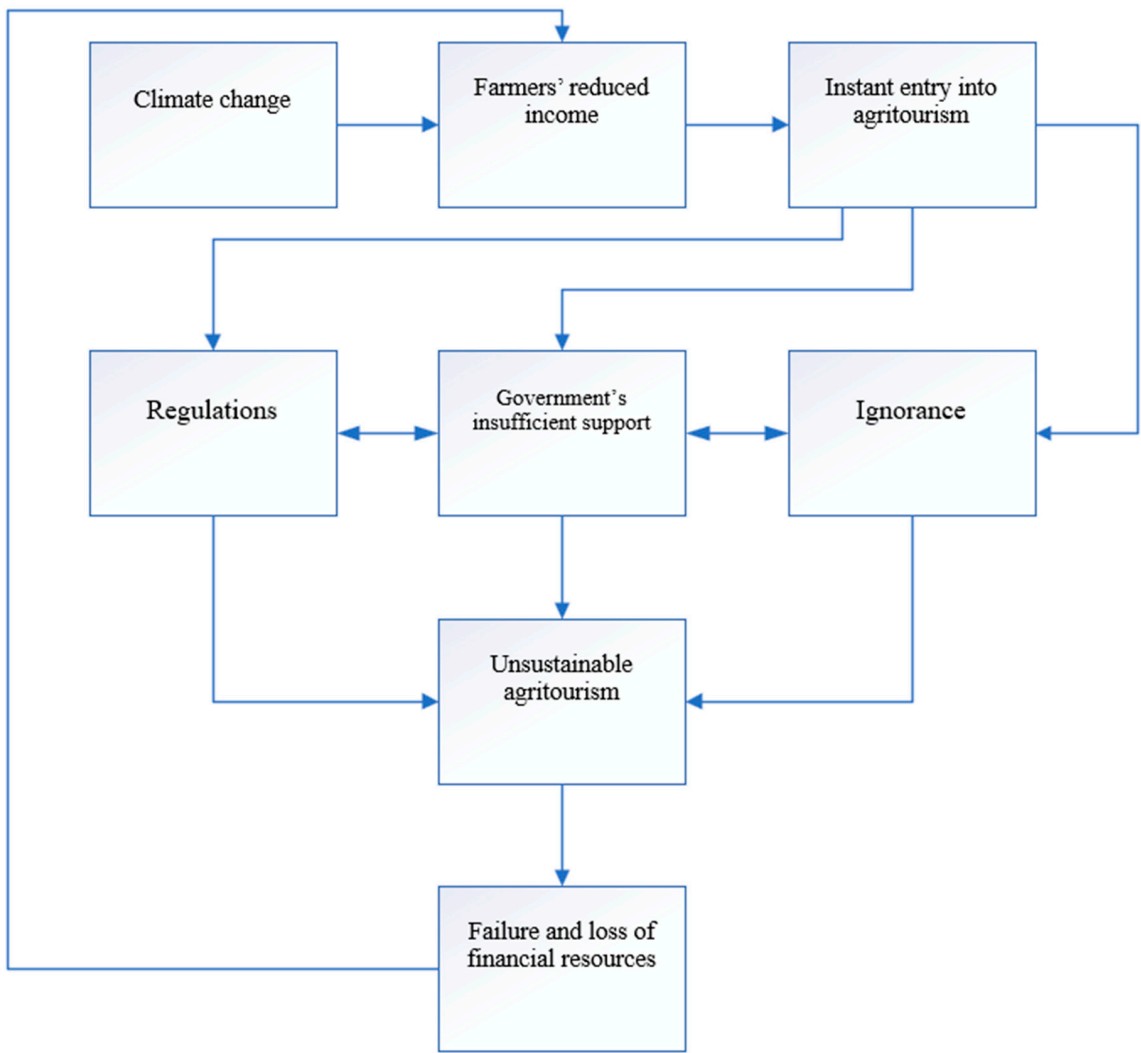

**Figure 2.** Mapping the landscape of agritourism in Shahrud, Iran. Illustration by authors.

## 6. Conclusions

By focusing on the unique negative effects of agricultural tourism development plans in the context of climate change adaptation, this work has filled a significant theoretical gap. Therefore, the purpose of this study was to investigate agritourism development policies that could be used to adapt to climate changes in the rural areas of Shahrud. The research identified several challenges that have hindered the success of adaptation policies. One of the significant challenges was the absence of comprehensive government support. The development of sustainable agritourism in rural areas may be hindered by the fact that, despite the government's promotion of agritourism in rural regions, farmers did not receive enough support through empowerment programs, capacity building, and incentives for infrastructure restoration.

Government assistance, on the other hand, is even more vital in Iran because the country lacks a strong private sector or NGOs, with practically all economic and social activity being either governmental or government-oriented. As a result, without substantial financial and governance backing from the public sector, the chances of success for development goals in Iran are minimal.

Another significant challenge identified in the study was the inadequate training of farmers involved in agritourism activities. It was discovered that farmers frequently lack the abilities and standards necessary to carry out agritourism operations effectively. This highlights the necessity for effective training programs and regulation of agritourism activities to ensure that farmers have the necessary skills and meet minimum standards. Furthermore, the study identified the lack of regulation as a significant challenge. The existing regulations did not provide special protection to farmers or tourists, making agritourism activities challenging to classify. Due to this, the rights of farmers and tourists are disregarded when it comes to government services, insurance, associated regulations, and security concerns. It is recommended that clear definitions and regulations for agritourism be created in Iran's official documents to provide adequate protection and support for both farmers and tourists.

Finally, the study revealed that unsustainable agritourism practices resulted from a lack of new approaches and immediate entry into the sector. This demonstrates how the concepts of sustainable agritourism are incompatible with the country's existing agritourism practices. To address these challenges, the study recommends that the government provides financial, educational, and regulatory support to agritourism entrepreneurs, including capacity building and infrastructure incentives. Additionally, it is important to strike a balance between promoting agritourism and protecting the environment, authenticity, and local culture, while increasing agricultural productivity.

### 6.1. Theoretical Implications

This study, conducted in Iran, has the potential to make a significant contribution to the literature on climate change and agritourism. The research highlighted the importance of government support in promoting an entrepreneurial culture in agritourism, building capacity, and strengthening local communities, which is consistent with the existing literature [32,35,78]. The study also identified the negative impacts of emotional and hasty policies, which can make it exacerbate farmers' challenges in the face of climate change. Risk management programs and the development of individual, societal, and institutional capacities can enhance farmers' adaptation and prevent hegemonic policies; this finding is consistent with the existing literature [79,80].

Furthermore, the research revealed how unsustainable agritourism methods have a detrimental influence on improving product quality, protecting the environment, and changing how land is used. The study highlighted the importance of managing agritourism sustainably by restricting visitor numbers, implementing water conservation and waste management practices, and ensuring equitable treatment of local communities and workers, in line with sustainable agritourism and community resilience principles [81,82].

Finally, the study contributed to our understanding of how decisions and policies are made in the agriculture sector under challenging circumstances, like climate change, particularly in developing nations like Iran. Studies in developing countries can provide insights into these processes and help inform future policy interventions [58]. This is because developing countries are often more vulnerable to the impacts of climate change due to their limited resources and capacity for adaptation. Therefore, understanding the experiences and challenges faced by farmers and policymakers in these contexts can inform future policy interventions and improve resilience to the impacts of climate change.

### 6.2. Practical Implications

The findings of the study have several practical implications for promoting agritourism in Shahrud. Firstly, the study suggested that the government should provide more support to farmers in the form of empowerment programs, capacity building, and financial incentives for infrastructure improvements. This could include financial support, training programs, and technical assistance to help farmers in acquiring the necessary skills and knowledge required to engage in agritourism ventures. Secondly, the study highlighted the need for clear regulations that provide special support for farmers as service providers and tourists as service recipients. In order to provide governmental services, insurance, related rules, and security issues, this could entail defining agritourism in Iran's laws and official papers. Thirdly, the study emphasized the necessity of managing agritourism sustainably. This could involve setting limits on the number of visitors, implementing water conservation and waste management practices, and ensuring that the local communities and workers are treated equitably. By taking a holistic approach to managing agritourism, it is possible to maximize its potential benefits while reducing its negative impacts. The study emphasized the need for stronger government support, clearer regulations, and sustainable management practices to promote agritourism in Shahrud. These findings could have practical implications for promoting agritourism not only in Iran but also in other countries with similar economic and social structures.

### 6.3. Limitations and Suggestions for Further Research

The limitations of a study are important to consider, as they may affect the generalizability and validity of the findings. In the case of the current research, there were several limitations that should be addressed in future studies. One major limitation was the study's narrow geographical scope. Although efforts were made to recruit participants from several villages in the Shahrud area, their perspectives and experiences may not be representative of other rural areas in Iran. To address this limitation, future studies should aim to broaden their scope and include participants from other regions with diverse cultural backgrounds. This will provide a more comprehensive understanding of the challenges and opportunities associated with agritourism development in Iran.

Suggestions for further research could include more detailed empirical studies of the agritourism sector in Iran, including quantitative analysis of its economic impact and qualitative research on the experiences of farmers, tourists, and other stakeholders. Comparative studies with other countries could also help to identification of best practices and policy lessons that could be applied in the Iranian context. Finally, more research is needed on the potential of agritourism to promote sustainable development and community resilience in rural areas, as well as the challenges and trade-offs involved in achieving these goals.

**Author Contributions:** Software, Z.-A.T.; Formal analysis, Z.-A.T. and N.B.K.; Investigation, A.R.K.-G.; Resources, N.B.K.; Data curation, Z.-A.T., A.R.K.-G. and N.B.K.; Writing—original draft, Z.-A.T. and C.M.H.; Writing—review & editing, A.R.K.-G. and C.M.H.; Visualization, A.R.K.-G.; Project administration, Z.-A.T. All authors have read and agreed to the published version of the manuscript.

**Funding:** This research received no external funding.

**Data Availability Statement:** This research was conducted in the form of qualitative interviews. No objective data can be presented.

**Conflicts of Interest:** The authors declare no conflict of interest.

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
