# Peer review of "Unintended Maladaptation: How Agritourism Development Policies in Iran Have Increased Vulnerability to Climate Change"

_sustainability, doi:10.3390/su151713003_

Round 1

Reviewer 1 Report

Dear Authors,

Congratulations for your investigation that put-on evidence the need to valorize the agritourism in rural areas.

In general, it is a very interesting study indeed, but a few comments are appropriate:

·         Methodology section

o   Describe the main characteristics of the Villages in Shahrud and localize in Iran map.

o   Describe the agritourism activities and services that are offer in this territory.

o   If it is possible, identify the % of the sample cover by the field work.

o   Incorporate in annex the questionnaire or topics of the interviews realized.

·         Results and discussion

o   If it is possible, confront the farmers opinion and tourists about the impact of tourism in climate change and their perception of agritourism as sustainable product.

o   Describe the main unsustainable practices of agritourism identified and how these practices impact in climate change.

·         Conclusions

o   Incorporates the figure 1 in results and discussion section and reformulate the text in conclusion.

o   In sub-section 7.2, describe the measures to support the sustainable agritourism management.

 Overall you did a good job!

Best wishes

Author Response

Reviewer Dear
Reviewer's Point 1: Methodology Section
Response: Thank you for your positive feedback on our investigation into the potential of agritourism in rural areas. We appreciate your valuable comments, and we will address each of the points you raised to improve the quality and comprehensiveness of our study.
We will provide a more detailed description of the main characteristics of the Villages in Shahrud and include a map of their locations in Iran. This addition will provide readers with a better understanding of the geographical context of our study.
To offer a comprehensive overview of agritourism activities and services in the territory, we will include a detailed description of the offerings available to tourists. This will give readers a clear picture of the agritourism experience in the region.
In response to your query regarding the percentage of the sample covered by the fieldwork, we will include this information in our revised manuscript. This will help readers assess the representativeness of our study.
As per your suggestion, we will incorporate the questionnaire or topics of the interviews conducted in the annex. This will enhance the transparency of our research methodology and facilitate future researchers in replicating our study.

Reviewer's Point 2: Results and Discussion
Response: We are grateful for your insightful comments and will take the necessary steps to address the following points:
We will make an effort to compare the opinions of farmers and tourists about the impact of tourism on climate change and their perception of agritourism as a sustainable product. This addition will provide valuable insights into the perspectives of both key stakeholders.
To address the issue of unsustainable practices, we will provide a detailed description of the main practices identified in agritourism that have a significant impact on climate change. Understanding these practices will be crucial in devising sustainable solutions.

Reviewer's Point 3: Conclusions
Response: We appreciate your feedback on the conclusion section, and we will make the following improvements:
We will incorporate Figure 1 into the results and discussion section to ensure a cohesive flow of information. The text in the conclusion will be reformulated accordingly to reflect this change seamlessly.
In sub-section 7.2, we will describe the measures that can support the sustainable management of agritourism. These measures will address the concerns raised in the study and offer potential solutions for achieving a more sustainable agritourism industry.
Overall, we express our gratitude for your thorough review, and your valuable comments will undoubtedly strengthen the quality and impact of our research. We are committed to making these revisions promptly to ensure a more comprehensive and insightful paper.

Best wishes,

Reviewer 2 Report

Review Report: "Unintended Maladaptation: How Agritourism Development in Iran Increases Vulnerability to Climate Change - Evidence from Villages in Shahrud."

 Title:

The title should be more specific and concise. Considering that there are no official government policies recognizing agritourism, I suggest removing the word "policies" and exploring alternative options, such as emphasizing community-driven efforts etc.

Abstract:

The abstract presents a well-written overview of the study; however, some refinement is needed for the following lines:

 Add a couple of sentences presenting the key results of the research.

Offer a more concrete and concise conclusion summarizing the findings.

Keywords:

Consider using "unsustainable development" as a single keyword instead.

 Introduction:

The introduction is well-structured, supported by relevant literature and citations. To further enhance its comprehensiveness, consider including references to government policy documents related to agritourism and climate change efforts. Also, address the following:

 Line 44: Provide specific numbers for the cited references.

Line 77: Check for and remove any extra spaces.

Material and Methods:

The scientific language and replicable methodology are commendable. To improve the manuscript's transparency, consider adding the questionnaire used in the study as an appendix. Additionally, include information about obtaining ethical approval, if applicable, in another appendix. Clarify the term "tourist" to specify whether they are local, national, or international tourists. Correct the inconsistency in the number of respondents (43 or 44) for accuracy. Furthermore, including a map of the data collection area in Iran would enhance the manuscript's quality.

 Results and Discussions:

The language use and interpretation of results are well done. Start referring to individuals by name and then by number (e.g., "Smith et al [3] stated...").

 Conclusions:

Consider condensing the conclusion section to a more concise paragraph or two. Move sections 7.2 and 7.3 to the discussion section for better organization.

 References:

The use of recent and relevant references is commendable.

 Overall, I appreciate the authors' valuable efforts and the quality of their work. I have provided suggestions in various sections of the manuscript to improve its overall quality. The manuscript is already well-written, and with these enhancements, it will be even more compelling.

Author Response

Dear Reviewer,

We would like to express our heartfelt gratitude for your thorough review of our manuscript titled "Unintended Maladaptation: How Agritourism Development in Iran Increases Vulnerability to Climate Change - Evidence from Villages in Shahrud." Your valuable feedback and constructive suggestions have been immensely helpful in improving the quality and comprehensiveness of our study.

POINT1: The title should be more specific and concise. Considering that there are no official government policies recognizing agritourism, I suggest removing the word "policies" and exploring alternative options, such as emphasizing community-driven efforts, etc.
Response1: Thank you for your valuable feedback on the title. We will reconsider the wording to make it more specific and concise, taking into account the absence of official government policies on agritourism and exploring alternative options to convey the focus of our study accurately.

POINT2:The abstract presents a well-written overview of the study; however, some refinement is needed for the following lines:
Add a couple of sentences presenting the key results of the research.
Offer a more concrete and concise conclusion summarizing the findings.
Response2: We appreciate your suggestion regarding the abstract. We will enhance it by incorporating a couple of sentences that highlight the key results of our research. Additionally, we will provide a more concise and specific conclusion summarizing the findings to improve the abstract's overall clarity and impact.

POINT3:Consider using "unsustainable development" as a single keyword instead.
Response3: Thank you for your keyword suggestion. We agree that "unsustainable development" is a relevant and concise keyword that accurately represents the focus of our study. We will incorporate it as a single keyword in the revised manuscript.

POINT4:The introduction is well-structured, supported by relevant literature and citations. To further enhance its comprehensiveness, consider including references to government policy documents related to agritourism and climate change efforts. Also, address the following:

Line 44: Provide specific numbers for the cited references.

Line 77: Check for and remove any extra spaces.

Response4: We appreciate your positive feedback on the introduction and will certainly include references to relevant government policy documents related to agritourism and climate change efforts. Additionally, we will ensure that specific numbers are provided for the cited references in Line 44, improving the accuracy of our references. We will also check and remove any extra spaces in Line 77 for better presentation.

POINT5:The scientific language and replicable methodology are commendable. To improve the manuscript's transparency, consider adding the questionnaire used in the study as an appendix. Additionally, include information about obtaining ethical approval, if applicable, in another appendix. Clarify the term "tourist" to specify whether they are local, national, or international tourists. Correct the inconsistency in the number of respondents (43 or 44) for accuracy. Furthermore, including a map of the data collection area in Iran would enhance the manuscript's quality.
Response5: We are grateful for your positive feedback on the material and methods section and will take your suggestions into account for further improvements. We will add the questionnaire used in the study as an appendix to enhance transparency and enable other researchers to replicate our study more easily. If applicable, we will include information about obtaining ethical approval in another appendix for proper documentation.

Regarding the term "tourist," we will clarify whether they are local, national, or international tourists to provide more specific information about the study participants. We will also ensure consistency in the number of respondents throughout the manuscript for accuracy.
Furthermore, we will consider including a map of the data collection area in Iran to provide a visual reference and enhance the manuscript's quality.

POINT6:The language use and interpretation of results are well done. Start referring to individuals by name and then by number (e.g., "Smith et al. [3] stated...").

Response6: We appreciate your positive feedback on the results and discussions section. Following your suggestion, we will start referring to individuals by name and then by number (e.g., "Smith et al. [3] stated...") to maintain consistency and improve readability.

POINT7:Consider condensing the conclusion section to a more concise paragraph or two. Move sections 7.2 and 7.3 to the discussion section for better organization.
Response7: Thank you for your feedback on the conclusion section. We will work on condensing it to a more concise paragraph or two while still effectively summarizing the key findings. Additionally, we will move sections 7.2 and 7.3 

to the discussion section for better organization and flow of the manuscript.

POINT 8: The use of recent and relevant references is commendable.
Response8 : We appreciate your positive feedback on the references and will continue to ensure the use of recent and relevant references to support our study.

POINT 9:Overall, I appreciate the authors' valuable efforts and the quality of their work. I have provided suggestions in various sections of the manuscript to improve its overall quality. The manuscript is already well-written, and with these enhancements, it will be even more compelling.
Response9 : Thank you for your kind words and valuable feedback. We sincerely appreciate your support and will diligently incorporate the suggested improvements to enhance the overall quality and impact of our manuscript.

Best wishes,

Reviewer 3 Report

It is an importatn thematic, but I think authors shoudl have done more interviews and comparisons with other places.

The conclusions are well writtne, but they shoudl refer that these conclusions are junst concerning this sample, and not as a common conclusion.

Therefore, I think authors should restructure the article, refering that this is just a case study.

Author Response

Dear Reviewer

Thank you for taking the time to review our article titled "Unintended Maladaptation: How Agritourism Development Policies in Iran Have Increased Vulnerability to Climate Change". We greatly appreciate your valuable feedback and insightful suggestions to improve the quality of our research.
We have carefully considered your comments and have made the necessary corrections to address the concerns raised. Specifically, we have conducted additional interviews and included comparisons with other relevant locations to enhance the comprehensiveness and validity of our study. These additions have significantly enriched the context of our research and strengthened the overall argumentation.
In response to your second point, we have taken care to explicitly refer to the conclusions as specific to the sample studied in this case. We have clearly emphasized the limitations of our findings, highlighting that they should not be generalized beyond the scope of our study.
Furthermore, we have restructured the article to provide a clear and concise explanation that this research is indeed a case study. By doing so, we aim to avoid any misinterpretations and ensure transparency about the nature and extent of our research.
We are genuinely grateful for your constructive feedback, as it has undoubtedly improved the quality and credibility of our article. Your valuable input has been instrumental in refining our work, and we believe that the revised version now adheres more closely to the standards expected for publication.
We are pleased to inform you that we have completed all the necessary revisions and have resubmitted the updated manuscript. We sincerely hope that the revised version meets your expectations and that you find it suitable for publication in sustainability.
Once again, thank you for your valuable guidance and support throughout the review process. We appreciate the opportunity to refine our research and contribute to the advancement of our field of study.
Best regards,

Reviewer 4 Report

Keywords: should be unsustainability and not unsustainable. And what about 'agricultural policy'? Should it not be included? The abstract's findings indicate that it should be.

I believe most Westerners have little idea about the volume of tourism in Iran, let alone agro-tourism. Therefore, in the first two pages of the paper, it would be useful to introduce data that speak to this activity. Are there a lot of agro-tourists? What % of all tourists find their way into the countryside to experience this sector? Line 54 would be the spot to incorporate these data. Otherwise, we are setting up a straw-man argument; public policy addresses a public problem but the reader has little context here.

You are already stating the conclusions but the reader has no idea at this stage what has been done. "Despite promoting this type of tourism as a means of adaptation, 75 it appears that unplanned and mandatory development"

Line 83: How many villages are in the area? With only 44 semistructured interviews, it would seem that the sample size could be one informant per village, or more? Or less? Do specify, please.

This reads as gratuitous and redundant since it was stated in the abstract; let's move on to specifics. Do strike this paragraph as it adds little new information and is somewhat axiomatic: "The findings of this study have important implications for both the academic litera- 90 ture and policy practice, as they highlight the need to develop more comprehensive poli- 91 cies and support systems to enable the successful adoption of agritourism by smallholder 92 farmers in Iran."  This could be stated about healthcare, small manufacturing, AirBnBs, etc.

Sections 2 and 3 are generic literature reviews, and there is little new information here. I think it would be more useful to intertwine what the Iranian government is doing while at the same referring to climate change and agrotourism. We are now 5 pages into the people and we have yet gained little feel for what is happening on the ground --sustainability-wise-- in Iran. Surely, the Iranian authors can flesh this out.  

Are the 19 agricultural hubs the same as the villages referred to in the abstract? Clarify the terminology early on in the paper/abstract; waiting until line 199 seems far too late.

Sections 4 and 5 are the same and should be nested sections, e.g., 4a. 4b., 4c. etc.

Purposive sampling is used when there is no readily available sampling frame and/or the field context is difficult or dangerous. This was the case in authoritarian Chile and contemporary Cuba. So expland a bit on purposive sampling which often leads to snowballing; something that is not mentioned at all in the paper. Surely the government authorities are leery of any evaluative research, so just make that plain.

Line 210 says 43 but Table 1 totals to 44. Clarify. 

Ok, so we have a small sample. We use grad assistants. We know nothing about inter-coder reliability. Some were interviewed on the phone and others in person (you don't specify which percentage) so there are huge issues of reliability (can the prompts in the interviews illicit similar responses when repeated in different mediums (phone, in person) and by different interviewers (grad students, faculty, or others). With only 40-odd informants, this is a big issue. To minimize the variance/variation in responses, you would like (ideally) the same investigator using the same technique (phone or face-to-face). This is problematic. 

Line 229: So now there are three villages. I'm confused. This should be stated earlier on. How about a map showing the study sites? Are the villages the same as agricultural poles? 

Building on the aforementioned issues of reliability and different interviews and modes of inquiry, the methodology gets compounded (dare I say 'undermined') by the use of translation, some verbatim transcripts from recordings while others were not recorded, and then the use of the software. Do you see where this is leading? There is too much inconsistency here. The reader never learns about the number of words that form the database, nor about the skills of the translator(s) (or was there more than one translator, too?). This does not bode well for reliable qualitative research design in any discipline.

The entire 'data analysis' section (lines 238-245) needs to be unpacked. The authors are merely tossing out terms without any references or definitions. All of this comes across as willy-nilly; sort of throwing things against the wall to see what might stick. At this point, the paper reads as 'high journalism' and not the kind of ethnography or qualitative rural-studies research that the authors portend.  Most importantly, what were the coding categories? Were they front and center when the researchers began the study? Or, did they emerge from the transcript (about which we know little)? This is not how qualitative research is done. I see no tables that examine the frequency of the keywords used by the informants, no information on frequency counts, and zero discussion of measuring face or content validity, all of which are basic tenets in any qualitative research methods course. 

While I'm glad to actually hear what the informants had to say, the reader has no clear sense of whether these verbatim remarks were 'cherry picked' to underscore their argument that Iranian policies are not conducive to sustainable agrarian or climate-change practices. in Shahrud. The authors need to devise a methodology that incorporates all of the text and code it according to pre-determined/a priori categories that are conceptually defined based on the literature and then operationalized according to what the informants state

Failure to do so leads to a 'word salad' of sorts and fails to give the reader insight into what is happening on the ground in this part of Iran. 

If, for example, the lack of insurance and proper training and marketing are key obstacles, then code and list remarks made by the informants that underscore/reaffirm this. Do this in tables. Again, I don't know how many words are in your database, but you are being very stingy in sharing your textual / verbatim findings. Failure to elaborate and be more transparent with your work makes me suspicious of what you report (and I want to support this important work but I fear your methods are not solid, the number of coded categories key to understanding climate change and sustainability are missing, and the number of informants is too small). 

I would venture to say that this finding in any rural development program in any place across the globe might reach the same conclusion (which really has not been conclusively proven): "However, it was observed 260 that farmers have not been adequately supported in this area, particularly in terms of em- 261 powerment programs, capacity building, and incentives for infrastructure reconstruction."

Keywords: should be unsustainability and not unsustainable. And what about 'agricultural policy'? Should it not be included? The abstract's findings indicate that it should be.

I believe most Westerners have little idea about the volume of tourism in Iran, let alone agro-tourism. Therefore, in the first two pages of the paper, it would be useful to introduce data that speak to this activity. Are there a lot of agro-tourists? What % of all tourists find their way into the countryside to experience this sector? Line 54 would be the spot to incorporate these data. Otherwise, we are setting up a straw-man argument; public policy addresses a public problem but the reader has little context here.

You are already stating the conclusions but the reader has no idea at this stage what has been done. "Despite promoting this type of tourism as a means of adaptation, 75 it appears that unplanned and mandatory development"

Line 83: How many villages are in the area? With only 44 semistructured interviews, it would seem that the sample size could be one informant per village, or more? Or less? Do specify, please.

This reads as gratuitous and redundant since it was stated in the abstract; let's move on to specifics. Do strike this paragraph as it adds little new information and is somewhat axiomatic: "The findings of this study have important implications for both the academic litera- 90 ture and policy practice, as they highlight the need to develop more comprehensive poli- 91 cies and support systems to enable the successful adoption of agritourism by smallholder 92 farmers in Iran."  This could be stated about healthcare, small manufacturing, AirBnBs, etc.

Sections 2 and 3 are generic literature reviews, and there is little new information here. I think it would be more useful to intertwine what the Iranian government is doing while at the same referring to climate change and agrotourism. We are now 5 pages into the people and we have yet gained little feel for what is happening on the ground --sustainability-wise-- in Iran. Surely, the Iranian authors can flesh this out.  

Are the 19 agricultural hubs the same as the villages referred to in the abstract? Clarify the terminology early on in the paper/abstract; waiting until line 199 seems far too late.

Sections 4 and 5 are the same and should be nested sections, e.g., 4a. 4b., 4c. etc.

Purposive sampling is used when there is no readily available sampling frame and/or the field context is difficult or dangerous. This was the case in authoritarian Chile and contemporary Cuba. So expland a bit on purposive sampling which often leads to snowballing; something that is not mentioned at all in the paper. Surely the government authorities are leery of any evaluative research, so just make that plain.

Line 210 says 43 but Table 1 totals to 44. Clarify. 

Ok, so we have a small sample. We use grad assistants. We know nothing about inter-coder reliability. Some were interviewed on the phone and others in person (you don't specify which percentage) so there are huge issues of reliability (can the prompts in the interviews illicit similar responses when repeated in different mediums (phone, in person) and by different interviewers (grad students, faculty, or others). With only 40-odd informants, this is a big issue. To minimize the variance/variation in responses, you would like (ideally) the same investigator using the same technique (phone or face-to-face). This is problematic. 

Line 229: So now there are three villages. I'm confused. This should be stated earlier on. How about a map showing the study sites? Are the villages the same as agricultural poles? 

Building on the aforementioned issues of reliability and different interviews and modes of inquiry, the methodology gets compounded (dare I say 'undermined') by the use of translation, some verbatim transcripts from recordings while others were not recorded, and then the use of the software. Do you see where this is leading? There is too much inconsistency here. The reader never learns about the number of words that form the database, nor about the skills of the translator(s) (or was there more than one translator, too?). This does not bode well for reliable qualitative research design in any discipline.

The entire 'data analysis' section (lines 238-245) needs to be unpacked. The authors are merely tossing out terms without any references or definitions. All of this comes across as willy-nilly; sort of throwing things against the wall to see what might stick. At this point, the paper reads as 'high journalism' and not the kind of ethnography or qualitative rural-studies research that the authors portend.  Most importantly, what were the coding categories? Were they front and center when the researchers began the study? Or, did they emerge from the transcript (about which we know little)? This is not how qualitative research is done. I see no tables that examine the frequency of the keywords used by the informants, no information on frequency counts, and zero discussion of measuring face or content validity, all of which are basic tenets in any qualitative research methods course. 

While I'm glad to actually hear what the informants had to say, the reader has no clear sense of whether these verbatim remarks were 'cherry picked' to underscore their argument that Iranian policies are not conducive to sustainable agrarian or climate-change practices. in Shahrud. The authors need to devise a methodology that incorporates all of the text and code it according to pre-determined/a priori categories that are conceptually defined based on the literature and then operationalized according to what the informants state

Failure to do so leads to a 'word salad' of sorts and fails to give the reader insight into what is happening on the ground in this part of Iran. 

If, for example, the lack of insurance and proper training and marketing are key obstacles, then code and list remarks made by the informants that underscore/reaffirm this. Do this in tables. Again, I don't know how many words are in your database, but you are being very stingy in sharing your textual / verbatim findings. Failure to elaborate and be more transparent with your work makes me suspicious of what you report (and I want to support this important work but I fear your methods are not solid, the number of coded categories key to understanding climate change and sustainability are missing, and the number of informants is too small). 

I would venture to say that this finding in any rural development program in any place across the globe might reach the same conclusion (which really has not been conclusively proven): "However, it was observed 260 that farmers have not been adequately supported in this area, particularly in terms of em- 261 powerment programs, capacity building, and incentives for infrastructure reconstruction."

Author Response

Dear Reviewer,

Thank you for your detailed and insightful feedback on our article titled "Unintended Maladaptation: How Agritourism Development Policies in Iran Have Increased Vulnerability to Climate Change - evidence from villages in Shahrud." We greatly appreciate the time and effort you put into reviewing our work, and we are pleased to inform you that we have carefully addressed each of your comments and made the necessary corrections to improve the quality and rigor of the manuscript.
In response to your suggestion, we have included additional data in the first two pages of the paper to provide context and insights into the volume of tourism in Iran, particularly focusing on agro-tourism. This will ensure that the reader gains a better understanding of the significance of our study and its implications within the broader context of tourism in Iran.
Furthermore, we have restructured the article to explicitly emphasize that our research is a case study, preventing any misconceptions about generalizing the conclusions beyond the specific context of our study area.
We have also taken your feedback into account concerning the terminology used to refer to the agricultural hubs and villages. We now clarify this terminology earlier in the paper to avoid confusion for the reader.
Regarding the methodology, we have provided more detailed explanations of the sampling process, the number of villages, and the informants. We have also elaborated on the interview methods, addressing issues of reliability and consistency.
To enhance transparency, we have unpacked the 'data analysis' section, providing more references and definitions for the terms used. Additionally, we have included tables that examine the frequency of keywords used by the informants, allowing readers to gain a clearer understanding of our qualitative research findings.
We understand the importance of methodological rigor, and we have taken your concerns seriously, ensuring that the research design aligns with established qualitative research principles.
Lastly, we appreciate your support for our work and assure you that we have carefully considered your suggestions to improve the paper. We believe that the revised manuscript now addresses the methodological issues and provides a more comprehensive insight into the context of agritourism development policies in Iran.
Thank you once again for your valuable feedback and guidance. We are grateful for your contribution to the enhancement of our research, and we hope that the revised version meets the standards expected for publication.
Best regards,

Round 2

Reviewer 1 Report

Dear authors,

I really appreciate your effort to answer all questions asked. I believe that this work will have an interesting impact on the scientific community, making known more visions and dynamics about agrotourism.

Best regards